# Towards Controllable and Interpretable Face Completion via Structure-Aware and Frequency-Oriented Attentive GANs

## Abstract

Face completion is a challenging conditional image synthesis task. This paper proposes controllable and interpretable high-resolution and fast face completion by learning generative adversarial networks (GANs) progressively from low resolution to high resolution. We present structure-aware and frequency-oriented attentive GANs. *The proposed structure-aware component* leverages off-the-shelf facial landmark detectors and proposes a simple yet effective method of integrating the detected landmarks in generative learning. It facilitates facial expression transfer together with facial attributes control, and helps regularize the structural consistency in progressive training. *The proposed frequency-oriented attentive module (FOAM)* encourages GANs to attend much more to finer details in the coarse-to-fine progressive training, thus enabling progressive attention to face structures. The learned FOAMs show a strong pattern of switching their attention from low-frequency to high-frequency signals. In experiments, the proposed method is tested on the CelebA-HQ benchmark. Experiment results show that our approach outperforms state-of-the-art face completion methods. The proposed method is also fast with mean inference time of $0.54$ seconds for images at $1024 \times 1024$ resolution (using a Titan Xp GPU).

## 1 Introduction

Conditional image synthesis aims to learn the underlying distribution governing the data generation with respect to the given conditions/context, which is also able to synthesize novel content. Much progress (Iizuka et al., 2017; Yeh et al., 2017; Li et al., 2017; Yang et al., 2016; Denton et al., 2016; Pathak et al., 2016; Yu et al., 2018; Liu et al., 2018; Brock et al., 2018; Karras et al., 2018) has been made since the generative adversarial networks (GANs) were proposed (Goodfellow et al., 2014). Despite the recent remarkable progress, learning controllable and interpretable GANs for high-fidelity image synthesis at high resolutions remain an open problem.

We are interested in controllable and interpretable GANs. We take a step forward by focusing on high-resolution and fast face completion tasks in this paper. Face completion is to replace target regions, either missing or unwanted, of face images with synthetic content so that the completed images look natural, realistic, and appealing. State-of-the-art face completion approaches using GANs largely focus on generating random realistic content. However, users may want to complete the missing parts with certain properties (e.g. expressions). Controllability is entailed. Existing face completion approaches are usually only able to complete faces at relatively low resolutions (e.g. $176 \times 216$ (Iizuka et al., 2017) and $256 \times 256$ (Yu et al., 2018)). To facilitate high-resolution image synthesis, the training methodology of growing GANs progressively (Karras et al., 2017) is widely used. For face completion tasks, one issue of applying progressive training is how to avoid distorting the learned coarse structures when the network is growing to a higher resolution. Interpretability is thus entailed to guide GANs in the coarse-to-fine pipeline. In addition, most existing approaches (Iizuka et al., 2017; Yeh et al., 2017; Li et al., 2017) require post-processing (e.g. Poisson Blending (Pérez et al., 2003)), complex inference process (e.g. thousands of optimization iterations (Yeh et al., 2017) or repeatedly feeding an incomplete image to CNNs at multiple scales (Yang et al., 2016)) during test.

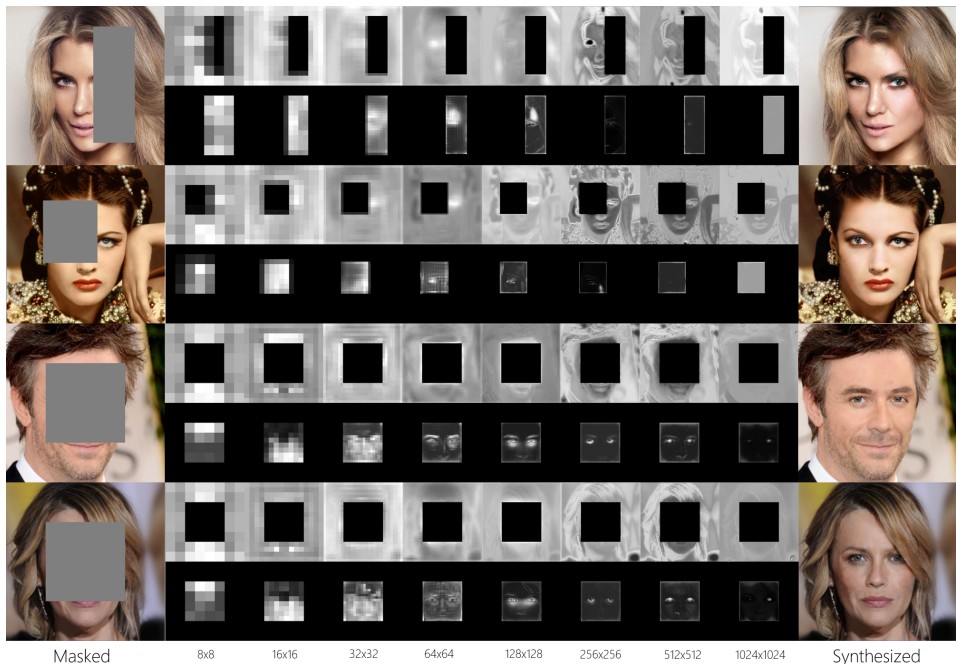

Masked  8x8  16x16  32x32  64x64  128x128  256x256  512x512  1024x1024  Synthesized

Figure 1: Face completion results of the proposed method on CelebA-HQ (Karras et al., 2017) at $1024 \times 1024$ resolution. The leftmost column are masked images while the rightmost are synthesized images. The learned FOAM filters are shown with higher intensities meaning more attention. At lower resolutions, the model focuses more on learning coarse structures (i.e. the lower-frequency signals). As the resolution increases, the model pays more attention to finer details (i.e. the higher-frequency information). Therefore, the FOAM partially and implicitly performs as a "band-pass filter" guiding the generation process. For instance, the model pays more attention to regions with richer details, such as hair and eyes, especially at high resolutions. The learned FOAM is also relatively stable when the target regions are similar, see the last two rows. Best viewed in color and magnification.

We present structure-aware and frequency-oriented attentive GANs that are progressively trained for high-resolution and fast face completion using a fast single forward step in inference without any post-processing. *By controllable,* it means that the completed face images can have different facial attributes (e.g., smiling vs not smiling) and/or facial expressions transferred from a given source actor. *By interpretable,* it means that the coarse-to-fine generation process in progressive training is rationalized. We utilize facial landmarks as backbone guidance of face structures and propose a straightforward method of integrating them in our system. We design a novel Frequency-Oriented Attention Module (FOAM) to induce the model to attend to finer details (i.e. higher-frequency content, see Fig. 1). We observe significant improvement of the completion quality by the FOAM against the exactly same system only without FOAM. A conditional version of our network is designed so that the appearance properties (e.g. male or female), and facial expressions of the synthesized faces can be controlled. Moreover, we design a set of loss functions inducing the network to blend the synthesized content with the contexts in a realistic way. Our method was compared with state-of-the-art approaches on a high-resolution face dataset CelebA-HQ (Karras et al., 2017). Both the evaluations and a pilot user study showed that our approach completed face images significantly more naturally than existing methods.

## 2 RELATED WORK

Recent learning based methods have shown the capability of CNNs to complete large missing content. Based on existing GANs, the Context Encoder (CE) (Pathak et al., 2016) encodes the contexts of masked images to latent representations, and then decodes them to natural content images, which are pasted into the original contexts for completion. However, the synthesized content of CE is often blurry and has inconsistent boundaries. Given a trained generative model, Yeh et al. (Yeh et al., 2017) propose a framework to find the most plausible latent representations of contexts to complete masked images. The Generative Face Completion model (GFC) (Li et al., 2017) and the Global and Local Consistent model (GL) (Iizuka et al., 2017) use both global and local discriminators, combined with post-processing, to complete images more coherently. Built on GL, Yu et al. (Yu et al.,

2018) design a contextual attention layer (CTX) to help the model borrow contextual information from distant locations. Liu et al. (Liu et al., 2018) incorporates partial convolutions to handle irregular masks. Unfortunately, these approaches can only complete face images in relatively low resolutions (e.g. $176 \times 216$ (Iizuka et al., 2017) and $256 \times 256$ (Yu et al., 2018)). Yang et al. (Yang et al., 2016) combine a global content network and a texture network, and the networks are trained at multiple scales repeatedly to complete high-resolution images ($512 \times 512$). But, they assume that the missing content always shares some similar textures with the context, which is improbable for the face completion task.

## 3 THE PROPOSED METHOD

### 3.1 PROBLEM FORMULATION

Denote by $\Lambda$ an image lattice (e.g., $1024 \times 1024$ pixels). Let $I_\Lambda$ be a face color image defined on the lattice $\Lambda$. Denote by $\Lambda_t$ and $\Lambda_{ctx}$ the target region to complete and the remaining context region respectively (note that the target region is not necessarily a single connected component, and the two parts form a partition of the lattice). $I_{\Lambda_t}$ is masked out with the same gray pixel value. Let $M_\Lambda$ be a binary mask image with all pixels in $M_{\Lambda_t}$ being 1 and pixels in $M_{\Lambda_{ctx}}$ being

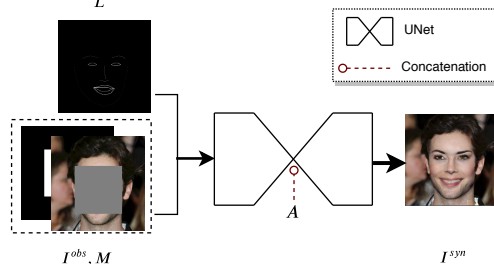

Figure 2: Overview of the proposed completion model. See text for details.

0. For simplicity, we will omit the subscripts $\Lambda$, $\Lambda_t$ and $\Lambda_{ctx}$ when the text context is clear. Unlike existing approaches (Pathak et al., 2016; Li et al., 2017; Iizuka et al., 2017) which first utilize unconditional image synthesis to generate the target region image and then blend them with context using using sophisticated post-processing, we address the completion problem as a coherent conditional image generation process.

As illustrated in Fig. 2, given an observed image $I^{obs}$ with the target region $I^{obs}_{\Lambda_t}$ masked out from a ground-truth uncorrupted image $I^{gt}$, **the objective of the proposed face completion** is to synthesize an image $I^{syn}$ that looks natural and realistic, and to enable a controllable generation process in terms of a given facial attribute vector, denoted by $A$ (such as male vs female, and smiling vs not smiling and for simplicity we use binary attribute vector in this paper) and/or a given facial expression encoded by facial landmark, denoted by $L$. Denote by $X^G = (I^{obs}, M, A, L)$ the input of the generator $G(\cdot)$ that realizes the completion. We have,

$$I^{syn} = G(X^G; \theta_G), \text{ subject to } I^{syn}_{\Lambda_{ctx}} \approx I^{obs}_{\Lambda_{ctx}}, \tag{1}$$

where $\theta_G$ collects all parameters of the generator and $\approx$ represents that the two context regions $I^{syn}_{\Lambda_{ctx}}$ and $I^{obs}_{\Lambda_{ctx}}$ need to be kept very similar.

**Structure-Aware Completion.** As illustrated in Fig. 3 (left), to enable transferring facial expressions in completion, we leverage the off-the-shelf state-of-the-art facial landmark detector, Face Alignment Network (FAN) (Bulat & Tzimiropoulos, 2017) which achieved very good results for faces in the wild. Motivated by this, we also want to integrate the landmark information in completion for faces without facial expression trans-

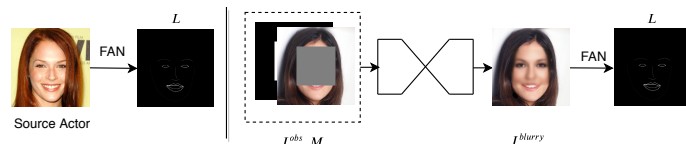

Figure 3: Illustration of computing facial landmarks for structure-aware completion. See text for detail.

fer required. Recent works (Isola et al., 2016; Wang et al., 2017; Zhu et al., 2017; Sangkloy et al., 2017; Xian et al., 2017; Chen & Hays, 2018) have shown the capability of GANs to translate sketches to photo-realistic images. We choose facial landmarks as an abstract representation of face structures in general. As illustrated in Fig. 3 (right), we first train a simple face completion model at the resolution of $256 \times 256$ using reconstruction loss (Section 3.3) only. Given an image [1], we use the trained model to generate a blurry completed image from which the landmarks are extracted with FAN (we observed that FAN can compute sufficiently good landmarks from blurry completed images). *Not only can this unify the generation process for different controllable settings (since the inputs to the generator are kept the same between with and without facial expression transfer),*

---

[1] The coarse completion model is only needed for testing. In training, we can extract landmarks from uncorrupted face images at the same resolution.

*but it also makes the completion process structure-aware.* Since faces have very regular structures (e.g. the eyes are always above a nose), when some facial components are occluded, it is possible to predict which parts are missing. Given a corrupted image, the quality of synthesized image can be further improved if the model is able to "draw" a sketch of the face first, which provides backbone guidance for image completion.

## 3.2 LEARNING WITH THE FOAM BETWEEN PROGRESSIVE STAGES

On top of GANS (Goodfellow et al., 2014), the framework of Context Encoder (CE) (Pathak et al., 2016) is adopted, so the generation process of our model is conditioned on the contextual information. The framework of training GANs progressively (Karras et al., 2017) is also adopted to facilitate a high-resolution completion model. This starts with the lowest resolution (such as $4 \times 4$). After running a certain number of iterations, higher resolution layers are added to both the generator and discriminator simultaneously until the network is grown to a desired resolution (such as $1024 \times 1024$). We present details of the proposed FOAM to stabilize and rationalize the progressive training.

Denote by $G_r$ and $D_r$ the generator and discriminator at a resolution level $r$, respectively, where $r \in \{1, \cdots, R\}$ is the index of resolution (e.g., $r = 1$ represents $4 \times 4$ and $r = R = 9$ represents $1024 \times 1024$). The final stage generator $G_R()$ will be used as the generator $G$ in Eqn. 1 in testing. The observed masked image, its corresponding binary mask, and the facial landmarks are re-sized to $I_r^{\text{obs}}$, $M_r$ and $L_r$ for each resolution respectively. In our model, both $G_r$ and $D_r$ are conditioned on facial landmarks. We attach the resolution index to the input and rewrite Eqn. 1 as,

$$I_r^{\text{syn}} = G_r(X^{G_r}; \Theta_{G_r}), \text{ subject to } I_{r, \Lambda_{\text{ctx}}}^{\text{syn}} \approx I_{r, \Lambda_{\text{ctx}}}^{\text{obs}}, \qquad (2)$$

where $X^{G_r} = (I_r^{\text{obs}}, M_r, A, L_r)$. For the discriminator $D_r$, its input is $X^{D_r} = (I_r, L_r)$ where $I_r$ represents either an uncorrupted image or a image synthesized by $G_r$. $D_r$ has two branches which share a common backbone and predict the *fake vs real* classification and the attribute estimation $\hat{A}$ respectively. The loss functions for training are defined in Section 3.3.

During progressive training, to avoid sudden changes to the trained parameters of $G_{r-1}$, the added layers (i.e. the higher resolution components) need to be faded into the networks smoothly during a *growing stage*. Since the parameters of added layers are initialized randomly, these layers may generate noise that distorts the coarser structures learned by $G_{r-1}$ if they are merged with $G_{r-1}$ directly. To reduce this effect, Karras et al. (Karras et al., 2017) use a linear combination of the higher and lower resolution branches. The synthesized image $\hat{I}^{\text{syn}}$ is computed by

$$\hat{I}^{\text{syn}} = \alpha I_r^{\text{syn}} + (1 - \alpha)\tilde{I}_{r-1}^{\text{syn}}, \qquad (3)$$

in which $I_r^{\text{syn}}$ and $\tilde{I}_{r-1}^{\text{syn}}$ are the output images from the higher and lower resolution branches respectively ($\tilde{I}_{r-1}^{\text{syn}}$ is up-sampled from $I_{r-1}^{\text{syn}}$ to match the resolution of $r$). $\alpha$ is a weight increasing linearly from zero to one during the growing stage. Therefore, at the beginning, the added layers have no impact on the network. During training, the influence of the higher resolution branch increases linearly while the weight of the lower-resolution branch decreases. In the end when $\alpha = 1$, the synthesized image depends only on the higher resolution branch (i.e. $\hat{I}^{\text{syn}} = I_r^{\text{syn}}$) and the lower

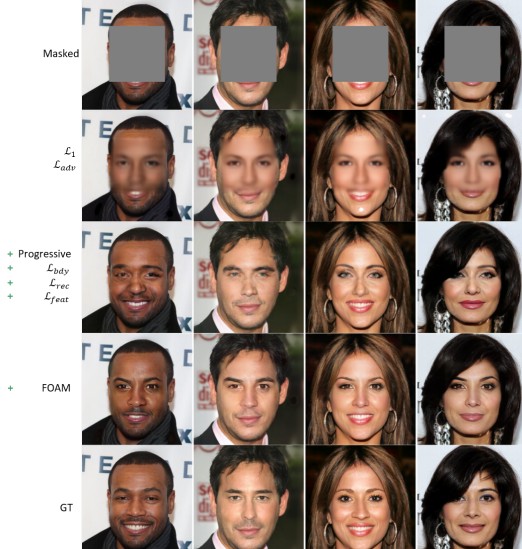

Figure 4: The ablation study shows the impact of essential components of our method. A model that is trained with adversarial $\mathcal{L}_{\text{adv}}$ and regular reconstruction loss $\mathcal{L}_1$ generates only blurry images. After adopting the progressive training method and a set of designed loss functions, the synthesized image quality is improved. By incorporating FOAM, the model focuses on learning only finer details while growing, resulting in sharper images with fewer distortions. Best viewed in magnification.

resolution branch can simply be removed. Because of this, once the training is complete, a cor-

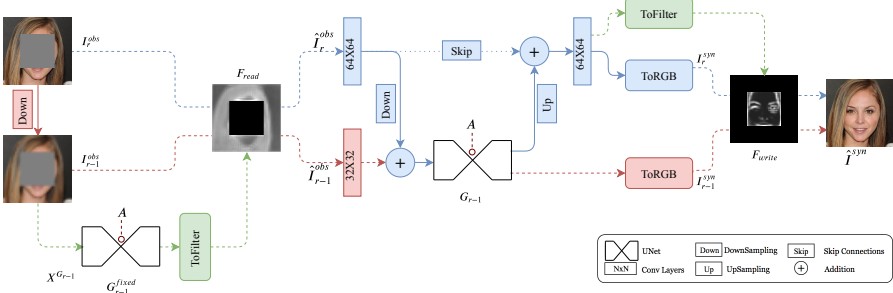

Figure 5: Illustration of the FOAM using an example of growing a $32 \times 32$ network to $64 \times 64$. The proposed FOAM consists of a *read* and a *write* operation to realize attentive "band-pass filters". See text for detail.

rupted image only needs to be fed to a single branch for image completion, and this process does not depend on any inputs or networks of lower resolutions.

**The FOAM.** Eqn. 3 is equivalent to applying "all-pass filters" to the higher and lower resolution branches, since all the pixels in images are assigned the same weight (i.e. $\alpha$ or $1 - \alpha$) regardless of their locations. Although this linear combination (Eqn. 3) has been shown effective for reducing the impact of noise generated during the growing stage, we observe that it does not work well for high-resolution face completion, as shown in Fig. 4. The coarse structures that have been learned well at lower resolutions are still vulnerable to being distorted during the joint training (i.e., $0 < \alpha < 1$).

*The intuitive idea of the proposed FOAM* is to encourage the generator to focus more on learning finer details during the growing stage, which is enabled by changing the "all-pass filters" reflected in Eqn. 3 to attentive "band-pass filters" that learn to protect what has been learned well in the previous stages and to update finer details as needed under the guidance of the loss functions. Existing approaches (Gregor et al., 2015; Yu et al., 2018) use spatial attention mechanisms to encourage networks to attend to selected parts of images (e.g. a rectangular region). As illustrated in Fig. 1, we observe that the FOAM filters indeed act like "band-pass filters" and show a strong pattern of switching its attention from coarse structures (i.e. the low-frequency information) to finer details (i.e. the high-frequency information) as the resolution increases. But, we note that different from regular band-pass filters, the filters learned by FOAM are predicted based on image semantics through the objective function (see Equation 14). This makes them sensitive to locations inferred on-the-fly in a coarse-to-fine manner. For instance, the model learns to pay more attention to eye regions where the rich details aggregate, especially at high resolutions. With the help of FOAM, the model is capable of learning meaningful and interpretable filters automatically.

As illustrated in Fig. 5, the proposed FOAM consists of a *read* and a *write* operation. In the ***read*** operation, only information that is important in $I_r^{\text{obs}}$ but does not exist in $I_{r-1}^{\text{obs}}$ will be allowed to enter the network. Similarly, in the ***write*** operation, only when the added layers produce information that can help reduce the overall loss, will it be allowed to add to the synthesized image $\hat{I}^{\text{syn}}$. The *read* and *write* operations, which are like two gates in a circuit, are controlled by the *read* and *write* filters learned by our model, respectively (denoted by $F_{\text{read}}$ and $F_{\text{write}}$). $F_{\text{read}}$ is predicted from the lower resolution branch and computed by,

$$F_{\text{read}} = \textit{ToFilter}\,(G_{r-1}^{\text{fixed}}(X^{G_{r-1}})), \qquad (4)$$

using a trained generator $G_{r-1}^{\text{fixed}}$ with fixed weights and a small trainable network *ToFilter*. Similarly, $F_{\text{write}}$ is predicted from the last feature maps of the higher resolution branch. The value in the filters represents the weight. $F_{\text{read}}$ helps extract the most valuable information in the contexts of $I_r^{\text{obs}}$ and $I_{r-1}^{\text{obs}}$. The ***read*** operation is implemented by,

$$\hat{I}_r^{\text{obs}} = F_{\text{read}} \odot (1 - M_r) \odot I_r^{\text{obs}}, \quad \hat{I}_{r-1}^{\text{obs}} = \textit{Downsample}\,((1 - F_{\text{read}}) \odot (1 - M_r) \odot \tilde{I}_{r-1}^{\text{obs}}), \quad (5)$$

where $\odot$ denotes element-wise multiplication. $\tilde{I}_{r-1}^{\text{obs}}$ is up-sampled from $I_{r-1}^{\text{obs}}$ to match the resolution of level $r$. Similar to Eqn. 3, $F_{\text{read}}$ and $(1 - F_{\text{read}})$ are assigned to the higher and lower resolution branches, respectively. The *write* filter $F_{\text{write}}$ combines the outputs from two branches (i.e. $I_r^{\text{syn}}$ and the up-sampled $\tilde{I}_{r-1}^{\text{syn}}$) to generate the final completed image $\hat{I}_r^{\text{syn}}$. $F_{\text{write}}$ helps extract the most valuable information in the contexts of $I_r^{\text{syn}}$ and $\tilde{I}_{r-1}^{\text{syn}}$. The ***write*** operation is defined by,

$$\hat{I}_r^{\text{syn}} = (I_r^{\text{syn}} \cdot \alpha + \tilde{I}_{r-1}^{\text{syn}} \cdot (1 - \alpha)) \odot (1 - M_r) + (F_{\text{write}} \odot I_r^{\text{syn}} + (1 - F_{\text{write}}) \odot \tilde{I}_{r-1}^{\text{syn}}) \odot M_r, \quad (6)$$

so, only the target region of $\hat{I}_r^{\text{syn}}$ is controlled by $F_{\text{write}}$. The context region is a linear combination of the contexts of $I_r^{\text{syn}}$ and $\tilde{I}_{r-1}^{\text{syn}}$.

To facilitate fast face completion in testing, we further design transformation functions to adjust the value ranges of $F_{\text{read}}$ and $F_{\text{write}}$, so the lower resolution branches and FOAMs can both be safely removed when the growing process is done. Similar to the vanilla progressive training method, a testing image only needs to go through the final stage for completion. To that end, a transformation function (Eqn. 7) is used to adjust the upper and lower bounds of the dynamic value ranges of the read and write filters. For instance, the transformed $\hat{F}_{\text{read}}$ starts as an all-zero filter, is adjusted by a trainable *ToFilter* at the growing stages, and eventually increased to all ones. The transformed filters $\hat{F}_{\text{read}}$ and $\hat{F}_{\text{write}}$ are defined by,

$$\hat{F}_{\text{read}} = \beta \cdot F_{\text{read}} + \gamma, \quad \hat{F}_{\text{write}} = \beta \cdot F_{\text{write}} + \gamma, \tag{7}$$

where the parameters are computed by

$$\beta : \begin{cases} 2\alpha, \\ 2 - 2\alpha, \end{cases} \quad \gamma : \begin{cases} 0, & \alpha \le 0.5 \\ 2\alpha - 1, & 0.5 < \alpha \le 1.0 \end{cases} \tag{8}$$

in which $\alpha$ is a weight increasing linearly from zero to one proportional to the number of seen images during growing. Eqn. 7 will be actually used in the read operation, Eqn. 5 and the write operation, Eqn. 6.

### 3.3 Loss Functions

To induce high-fidelity face completion, we utilize the loss functions as follows.

**Adversarial Loss** Given an uncorrupted ground-truth image $I^{\text{gt}}$, its attribute vector $A$, a mask $M$, landmarks $L$, and the corresponding corrupted image $I^{\text{obs}}$, we define the loss by,

$$l_{\text{adv}}(I^{\text{gt}}, M, L, I^{\text{obs}}, A|G, D) = \log\left(1 - D_{\text{cls}}(I^{\text{syn}}, L) + \log D_{\text{cls}}(I^{\text{gt}}, L), \tag{9}$$

where $I^{\text{syn}} = G(I^{\text{obs}}, M, A, L)$ and $D_{\text{cls}}$ represents the classification branch of the discriminator.

**Attribute Loss** Similar to the InfoGAN models (Chen et al., 2016; Choi et al., 2017), for the attribute prediction head classifier in the discriminator, we define the attribute loss based on cross-entropy between the predicted attribute vectors, $\hat{A}^{\text{gt}} = D_{\text{attr}}(I^{\text{gt}}, L)$ and $\hat{A}^{\text{obs}} = D_{\text{attr}}(I^{\text{obs}}, L)$, and the corresponding input attribute vectors $A$ for both a ground-truth image and a synthesized image,

$$l_{\text{attr}}(I^{\text{gt}}, A, M, I^{\text{obs}}|G, D) = CrossEntropy\,(A, \hat{A}^{\text{gt}}) + CrossEntropy\,(A, \hat{A}^{\text{obs}}), \tag{10}$$

where $D_{\text{attr}}$ represents the attribute prediction branch of the discriminator.

**Reconstruction Loss** Since our method generates an entire completed face, we define a weighted reconstruction loss $l_{\text{rec}}$ to rebuild both the content and the context regions,

$$l_{\text{rec}}(I^{\text{gt}}, M, L, I^{\text{obs}}, A|G) = \|\kappa \odot M \odot I^{\text{diff}}\|_1 + \|(1 - \kappa) \odot (1 - M) \odot I^{\text{diff}}\|_1, \tag{11}$$

where $I^{\text{diff}} = I^{\text{gt}} - I^{\text{syn}}$ and $\kappa$ is the trade-off parameter.

**Feature Loss** In additional to the reconstruction loss, we also expect a synthesized image to have similar feature representations (Johnson et al., 2016) to a ground-truth image. Let $\phi$ be a pre-trained deep neural network and $\phi_j$ be the activation of the $j^{th}$ layer of $\phi$, the feature loss is defined by,

$$l_{\text{feat}}(I^{\text{gt}}, M, L, I^{\text{obs}}, A|\phi, G) = \|\phi_j(I^{\text{gt}}) - \phi_j(I^{\text{syn}}))\|_2^2. \tag{12}$$

In our experiments, $\phi_j$ is the *relu2_2* layer of a 16-layer VGG network (Simonyan & Zisserman, 2014) pre-trained on the ImageNet dataset (Russakovsky et al., 2015).

**Boundary Loss** To make the generator learn to blend the synthesized target region with the original context region seamlessly, we further define a close-up reconstruction loss along the boundary of the mask. Similar to (Yeh et al., 2017), we first create a weighted kernel $w$ based on the mask image $M$. $w$ is computed by blurring the mask boundary in $M$ with a mean filter so that the pixels closer to the mask boundary are assigned larger weights. The kernel size of the mean filters is seven in our experiments. We have,

$$l_{\text{bdy}}(I^{\text{gt}}, M, L, I^{\text{obs}}, A|G) = \|w \odot (I^{\text{gt}} - I^{\text{syn}})\|_1. \tag{13}$$

Our model is trained end-to-end by integrating the expected loss of the loss functions defined above under the minimax two-player game setting. The full objective is,

$$\min_G \max_D \mathcal{L}_{\text{adv}}(G, D) + \lambda_1 \mathcal{L}_{\text{attr}}(G, D) + \lambda_2 \mathcal{L}_{\text{rec}}(G) + \lambda_3 \mathcal{L}_{\text{feat}}(G, \phi) + \lambda_4 \mathcal{L}_{\text{bdy}}(G), \tag{14}$$

where $\lambda_i$'s are trade-off parameters between different loss terms. Fig. 4 shows an ablation study of the importance of the loss functions. Note that the ablation study was run at $256 \times 256$. Since the training of high-resolution models was very time consuming, an ablation study for $1024 \times 1024$ networks is left for future work.

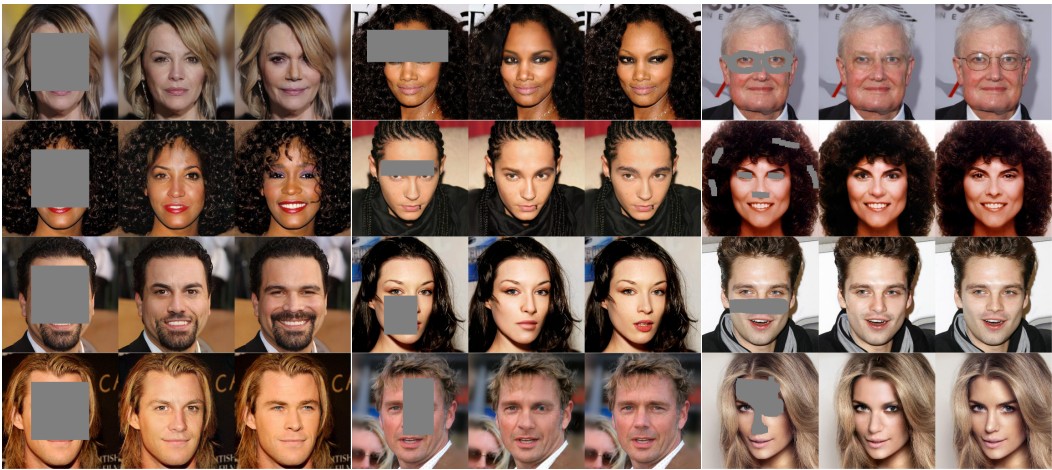

Figure 6: Examples of high-resolution face completion results by our method at $1024 \times 1024$ resolution. For each group, from left to right: masked, synthesized and real images. Our model was able to capture the anatomical structures of faces and handle various shaped masks.

## 4 EXPERIMENTS

**Datasets and Experiment Settings.** We used the CelebA-HQ (Karras et al., 2017) dataset for evaluation. It contains 30,000 aligned face images at $1024 \times 1024$ resolution. The dataset is split randomly **while ensuring there is no identity overlap between test/training sets:** 3,009 images for testing, and 26,991 for training. There were two types of masks in training: center and random. The center mask was a square region in the middle of the image with a side length of half the size of the image. The random masks, generated in a similar way to previous methods (Iizuka et al., 2017; Yu et al., 2018), were rectangular regions with random width-to-height ratios, sizes and locations covering about $5\%$ to $25\%$ of the original images. *Network architectures, hyper-parameters and more results are provided in the Appendix.*

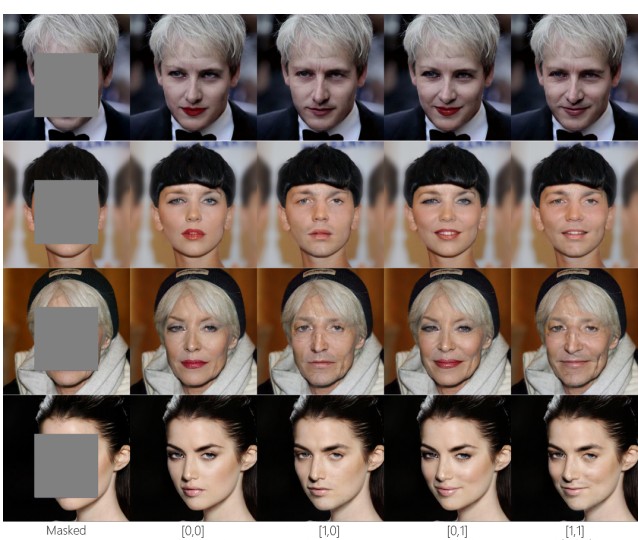

Figure 7: Examples of controlling attributes. All images are at $512 \times 512$ resolution. The leftmost column are masked images, and the rest are generated faces. More in Fig. 14 and Fig. 15.

**Completion *without* Attribute Control.** We first trained a high-resolution ($1024 \times 1024$) model with center masks (examples shown in Fig. 6) to test whether our model is capable of learning high-level semantics and structures of faces and synthesizing large missing regions. The second model was trained with random rectangular masks, but was able to handle various shaped masks (e.g. irregular hand-drawn masks). Fig. 6 shows that our model was able to capture the anatomical structures of faces and generate content that is consistent with the holistic semantics.

**Completion *with* Attribute Control.** Two attributes ("Male vs Female" and "Smiling vs Not Smiling") were chosen. This model was trained from scratch and the result was run at a $512 \times 512$ resolution. Fig. 7 shows that the attributes of synthesized images were controlled by our model explicitly. Fig. 8 shows the facial expression transfers together with attribute control.

**Quantitative Evaluation.** In current literature (Yeh et al., 2017; Yu et al., 2018), reconstruction metrics such as mean *L1*, *L2* errors and peak signal-to-noise ratio (PSNR) are commonly used for the evaluation of in-painting methods. We show the comparison between our method and state-

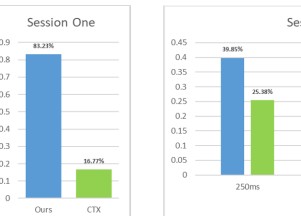
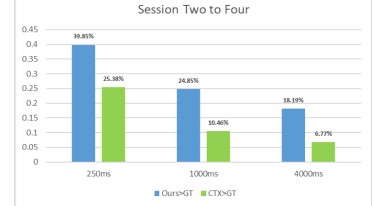

Figure 8: Examples of joint attribute and expression control. All images are at $512 \times 512$ resolution. Though the source and synthesized faces have different identities, their expressions are very similar.

of-the-art models at their reported resolutions respectively (Table 1). The result shows that our model outperformed state-of-the-art approaches. In addition to comparing how well each model reconstructs the missing regions, it is also important to evaluate the naturalness of synthesized images since image completion aims to generate realistic and plausible content rather than restoring the original images perfectly. Due to the lack of good metrics for naturalness, *we ran a pilot user study, which is considered the "gold standard" to evaluate GAN models. Our method obtains significantly better results (Fig. 9, see detail in the Appendix).*

Table 1: The quantitative comparison between our method and state-of-the-art methods

| Method | Resolution | L1 (%) | L2 (%) | PSNR |
|---|---|---|---|---|
| GL (Iizuka et al., 2017) | $128 \times 128$ | 9.34 | 1.75 | 18.22 |
| Ours | $128 \times 128$ | **7.8** | **1.42** | **19.15** |
| CTX (Yu et al., 2018) | $256 \times 256$ | 8.53 | 1.75 | 18.41 |
| Ours | $256 \times 256$ | **7.05** | **1.21** | **19.97** |

**Computation Time.** We tested our model with a Titan Xp GPU by processing 3000 $1024 \times 1024$ images with $512 \times 512$ holes. The mean completion time is 0.54 second per image. It takes about one minute for the model of Yang et al. (Yang et al., 2016) to complete a $512 \times 512$ scene image with a Titan X GPU.

**Limitations** Though our method has low inference time, the training time is long due to the progressive growing of networks. In our experiment, it takes about three weeks to train a $1024 \times 1024$ model on a Titan Xp GPU. By carefully zooming in our results, we find that our high-resolution model fails to learn low-level skin textures, such as furrows and sweat

*Figure 9: Comparisons on the naturalness: ours and CTX (Yu et al., 2018). Left: There was a significantly higher percentage of images completed by our model that looked more realistic than those completed by CTX. Right: The percentage that a synthesized image is considered more realistic than a ground-truth (GT) one. There is a significantly higher probability that images completed by our method were classified as real samples versus those generated by CTX.*

holes. Moreover, the model could generate distorted content when removing large parts (e.g. hats) or synthesize some plausible but unnatural faces (Fig. 10). Furthermore, for facial expression transfer, our method requires that the head poses of the source and target faces are similar. These issues are left for future work.

## 5 CONCLUSION

We propose a progressive GAN with frequency-oriented attention modules (FOAM) for high resolution and fast controllable and interpretable face completion, which learns face structures from coarse to fine guided by the FOAM. By consolidating information across all scales, our model not only outperforms state-of-the-art methods by generating sharper images in low resolution (such as $256 \times 256$), but is also able to synthesize faces in higher resolutions (such as $512 \times 512$ and $1024 \times 1024$) than existing techniques. Our attribute and expression controller allows users to manipulate the appearance and facial expressions of generated images explicitly with attribute vectors and landmarks. Our system is designed in an end-to-end manner, in that it learns to generate completed faces directly and more efficiently.

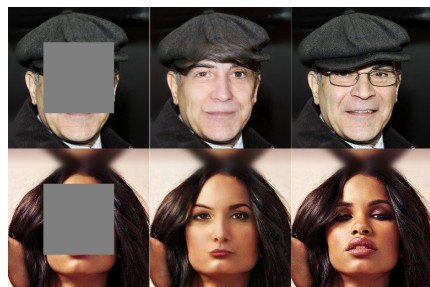

Figure 10: Some failure cases. From left to right: masked, synthesized, and real images.

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

## A APPENDIX

In this section, we first show more results of high resolution face completion. Then, we present detail of the user study. We also provide detail of the network architectures and hyper-parameters used in training.

### A.1 MORE RESULTS OF FACE COMPLETION

Fig. 11, Fig. 12 and Fig. 13 show more results of high resolution face completion with masks of various shape. Our model is capable of learning high-level semantics and structures of faces and handling challenging mask types that were not included in the training set (e.g. hand-drawn masks in Fig. 13).

Fig. 14 and Fig. 15 show more results of hard and soft attribute control respectively. Fig. 16 show more results of joint attribute and expression control.

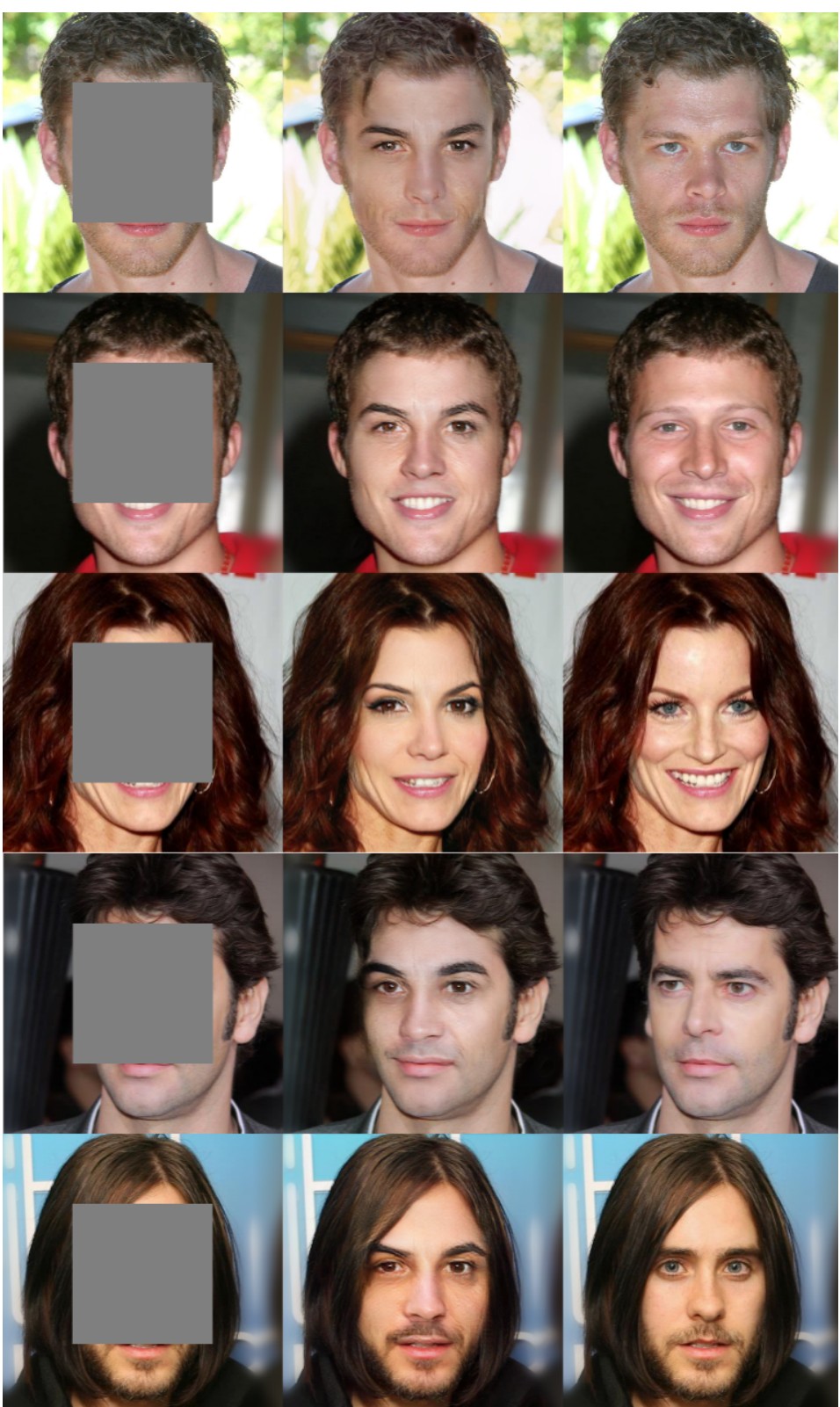

Figure 11: High-resolution face completion results with center masks. All images are at $1024 \times 1024$ resolution. For each group, from left to right: masked, synthesized, and real images.

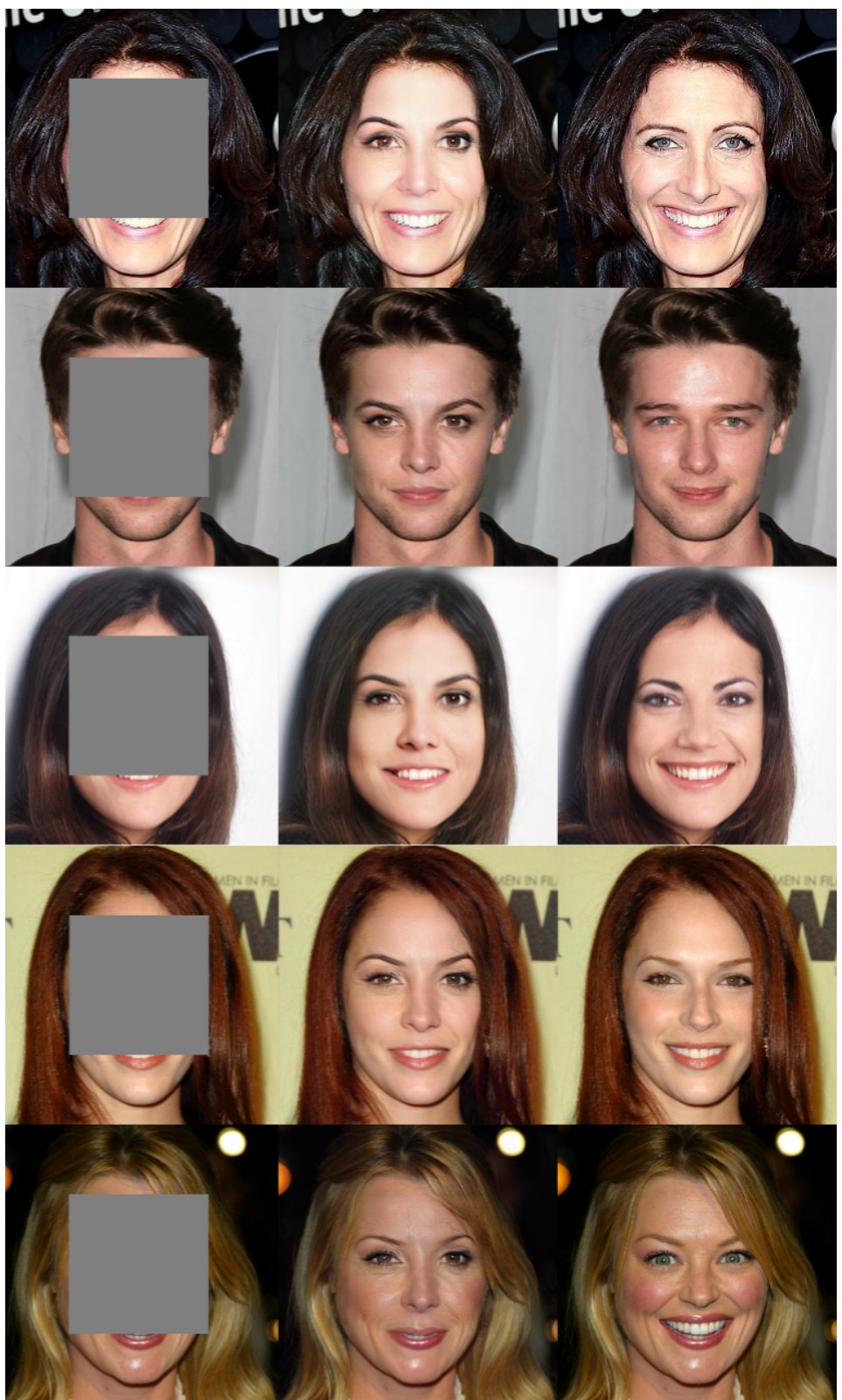

Figure 12: More examples of High-resolution face completion results with center masks. All images are at $1024 \times 1024$ resolution. For each group, from left to right: masked, synthesized, and real images.

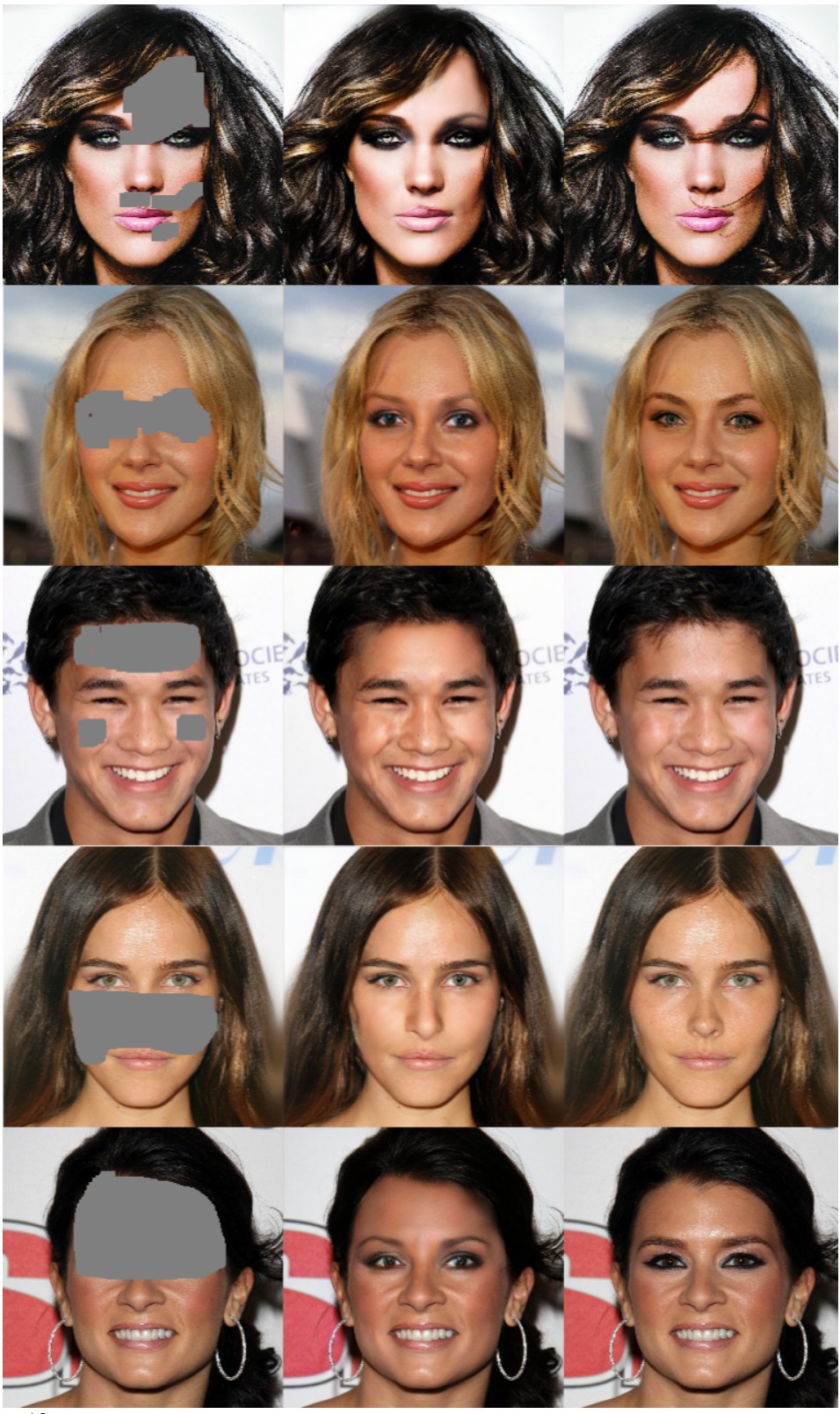

Figure 13: More examples of High-resolution face completion results with hand-drawn masks. All images are at $1024 \times 1024$ resolution. For each group, from left to right: masked, synthesized, and real images.

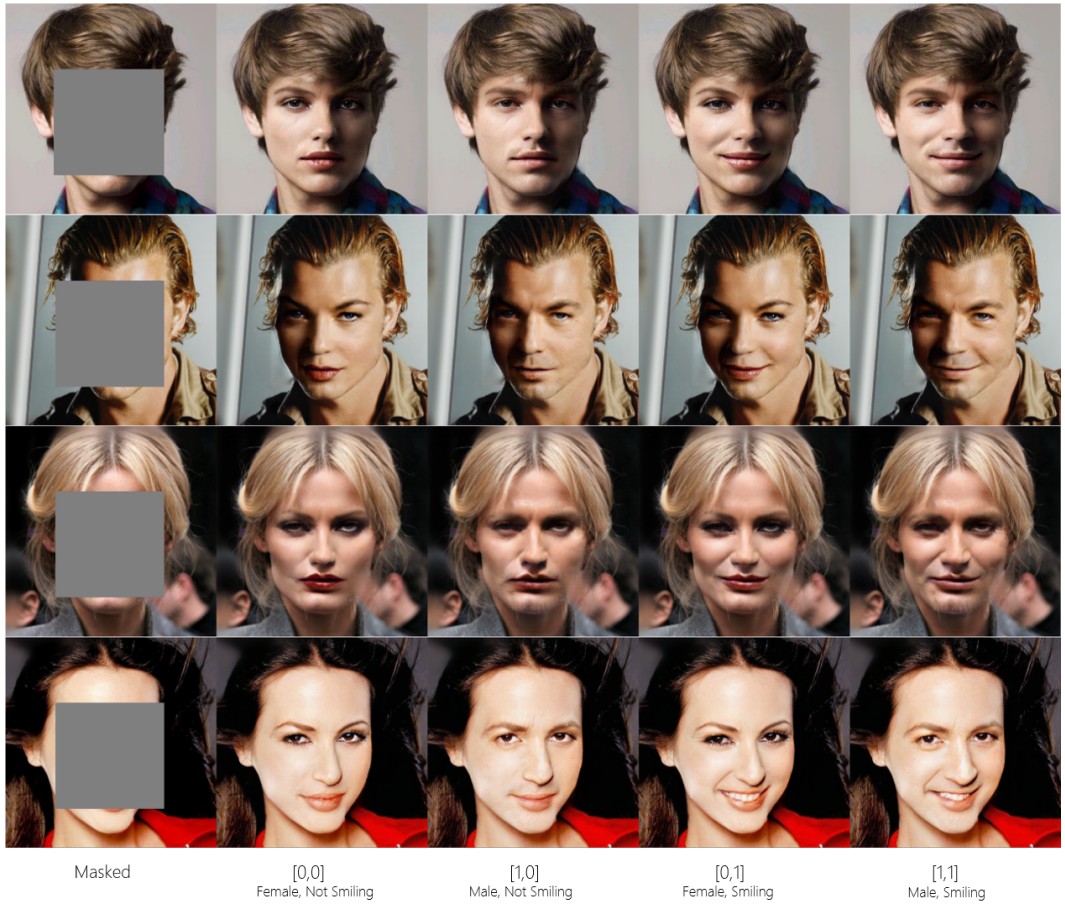

| Masked | [0,0]
Female, Not Smiling | [1,0]
Male, Not Smiling | [0,1]
Female, Smiling | [1,1]
Male, Smiling |

Figure 14: Face completion results with attribute controller. Attribute "Male vs Female" is used to control the appearance. Landmarks from source actors are used to control the facial expressions of synthesized images. The leftmost column shows masked images and faces generated with ground-truth attributes and landmarks.

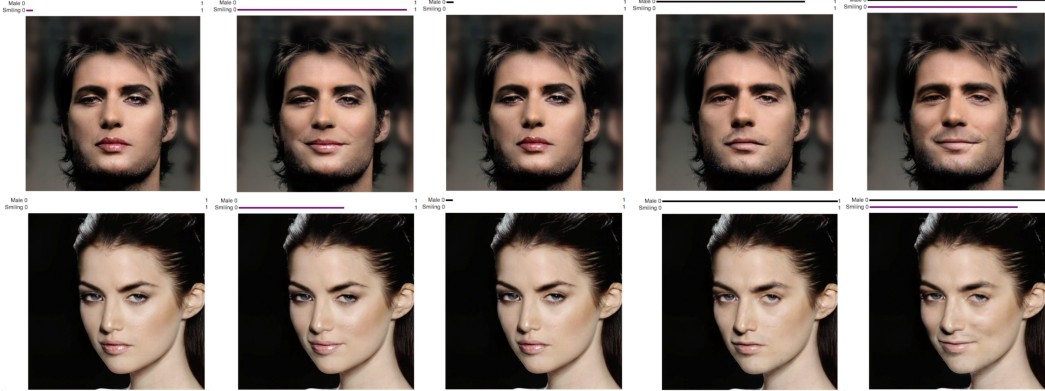

Figure 15: Snapshots of face completion results with **relative and soft attribute controller**. The Demo video will be available as supplementary material.

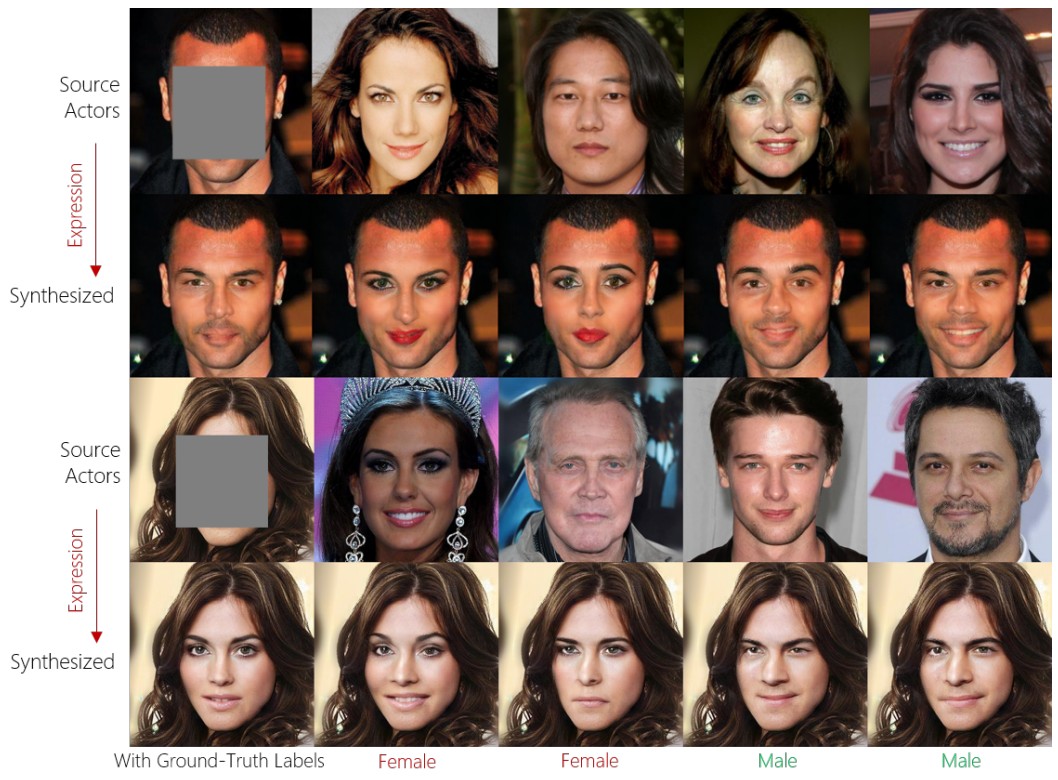

Figure 16: Face completion results with attribute controller. Attribute "Male vs Female" is used to control the appearance. Landmarks from source actors are used to control the facial expressions of synthesized images. The leftmost column shows masked images and faces generated with ground-truth attributes and landmarks.

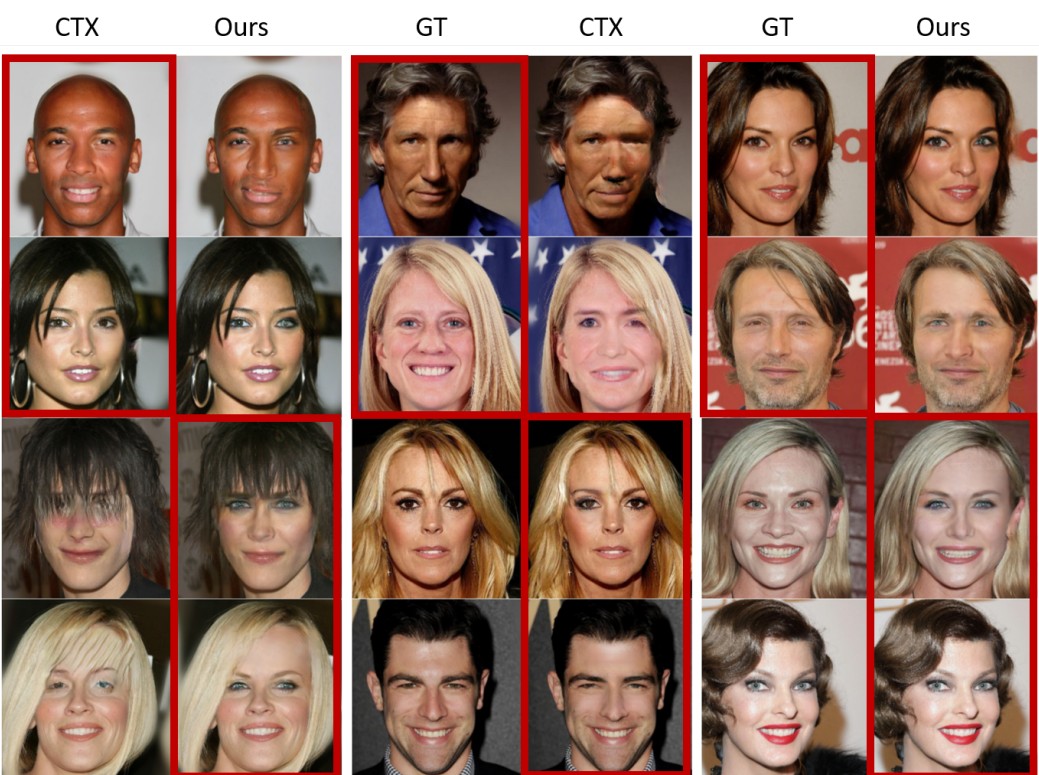

Figure 17: Examples of images used in the user study. The preferred images are marked with red boxes.

A.2 DETAIL OF USER STUDY

We compared our method with CTX (Yu et al., 2018), which is a state-of-the-art CNN-based face completion approach capable of completing face images at $256 \times 256$ resolution, with a pilot user study at $256 \times 256$ resolution with random masks. 27 subjects (15 male and 12 female participants, with ages from 22 to 32) volunteered to participate.

There were four sessions in the experiments. For each trial, a user was shown two images and asked to choose the more realistic one. In the first session, two images completed from the same image by different methods were chosen (one by our model and another by CTX). In sessions two to four, a real image and a corresponding synthesized image were shown. In the first session, time was unlimited. In session two to four, images were on display for 250ms, 1000ms, 4000ms respectively. We randomly chose 100 pairs of images [Ours, CTX] from the test set for session one and another 100 groups of images [Ours, CTX, GT] for session two to four. The display order of images was randomized.

The result (Fig. 9) shows that there was a significantly higher chance that images completed by our model looked more realistic than those completed by CTX in session one. While comparing with the ground-truth images in session two to four, the highest possible percentage that a method can achieve is 50%, which is a random guessing. There was a probability of about 40% that our method could fool a human observer when the display time was 250 ms. As the display time increased, users started to notice more detailed artifacts, so all the percentages dropped. Regardless, our model always significantly outperformed CTX. Statistical analysis was performed to confirm the significant differences between our method and CTX. The details of analysis are listed in the supplemental materials.

Some of the most-frequently-picked images by participants are shown in Fig. 17. Overall, our approach generated sharper images with more details and fewer distortions. Sometimes, the synthesized faces looked even more natural than the ground-truth images.

Table 2: The results of the two-way repeated measures ANOVA of the user study. There was a strong main effect for Method, which indicated that the images generated by our method were recognized as real ones by the human observers significantly more frequently than those completed by CTX (Yu et al., 2018).

| Source | F | p |
|---|---|---|
| Method | F(1,26) = 352.645 | **p<0.001** |
| Time | F(1.657,43.079) = 203.235 | **p<0.001** |
| Method×Time | F(1.283,33.346) = 1.760 | p=0.194 |

**Statistical Analysis of the User Study.** In order to confirm the intuition of our comparison results, we tested for statistical significance. To do this, we first collapsed each participant's rankings into a frequency list. Once frequency lists were built for all participants, the frequencies for each method were again averaged over the 27 participants to produce a final list of averages from $n = 27$ samples. For session one, we performed the paired samples t-test to compare the means of frequencies for these two methods: $t(26) = 46.368, p < 0.001$. The results confirmed that our method was favored significantly more often than CTX. For session two through four, a two-way repeated measures analysis of variance (ANOVA) was used because there were two factors: *Method* (ours and CTX) and *Time* (250 ms, 1000 ms, and 4000 ms). Since the sphericity assumption was not met in our data, we used the correction of Huynh-Feldt for the *Method* and *Time* factors, and the Greenhouse-Geisser correction for the *Method×Time* interactions. Not surprisingly, results (Table 2) showed a significant difference in means of different method groups for a standard $\alpha = 0.05$, which denoted that there was a significantly higher probability that images completed by our method were classified as real samples versus those generated by CTX.

A.3 NETWORK ARCHITECTURES AND HYPER-PARAMETERS

The generator $G$ in our model is implemented by a U-shape network architecture consisting of the first component $G_{enc}$ transforming the observed image and its mask to a latent vector and the second component $G_{dec}$ transforming the concatenated vector (latent code and input attributes) to a completed image. There are residual connections between layers in $G_{enc}$ and the counterpart in $G_{dec}$ similar in the spirit to the U-Net (Ronneberger et al., 2015) and the Hourglass network (Newell

et al., 2016) to consolidate information across multiple scales. Fig. 19 illustrates the two structures of a layer in the generator for training without and with the attribute controller respectively, which are adapted from the U-Net and Hourglass network.

Every convolutional layer (Conv) is followed by an Instance Normalization (InsNorm) and a LeakyReLU layer, except that the Conv before the latent vector (i.e. the second Conv layer in Table 3) is not followed by an InsNorm. Additionally, the there are no InsNorms or LeakyReLUs after the last Convs of both $D_{cls}$ and $D_{attr}$. All Convs used in the residual block of the skip connections of our conditional model have a kernel size of three and a stride of one.

Since we use Instance Normalization rather than Batch Normalization, the batch size is not an important hyper-parameter. Technically, for faster computation, we use as large a batch size as possible so long as it does not exceed the GPU memory limit.

Tables 3 and 5 demonstrate the architecture of the components of the generator $G$ while Tables 6 shows the components of the discriminator $D$. In Table 6, depending on the operation of the skip connection (Skip), the number of filters is either doubled (for a concatenation operation) or remains the same (for an addition operation).

The progressive training process is illustrated in Fig. 18. At a resolution lower than $1024 \times 1024$, the input face images, masks, landmarks and real images are all down-sampled with average pooling to fit the given scale. One of the major challenges of generating high resolution images is the limitation of Graphics Processing Unit (GPU) memory. Most completion networks use Batch Normalization (Ioffe & Szegedy, 2015) to avoid covariate shift. However, with the limited GPU memory, only a small number of batch sizes are supported at high resolution, resulting in low quality of generated images. We use the Instance Normalization (Ulyanov et al., 2016), similar to Zhu et al. (Zhu et al., 2017), and update $D$ with a history of completed images instead of the latest generated one (Shrivastava et al., 2016) to stabilize training.

At the growing stage, new layers are added for both $D$ and $G$ and these layers are faded in with current networks smoothly. After the fade-in process, the network is trained on more images for stabilization. We used 300K, and 150K training images for resolution [$8 \times 8$ to $256 \times 256$] and [$512 \times 512$, $1024 \times 1024$] respectively at growing stage, and 600K, 430K images for $4 \times 4$ and [$8 \times 8$ to $1024 \times 1024$] at stabilizing stage respectively.

In the experiments, the reconstruction trade-off parameter was set to $\kappa = 0.7$ to focus more on the target region. To balance the effects of different objective functions, we used $\lambda_{attr} = 2$, $\lambda_{rec} = 500$, $\lambda_{feat} = 8$, and $\lambda_{bdy} = 5000$. The Adam solver (Kingma & Ba, 2014) was employed with a learning rate of 0.0001.

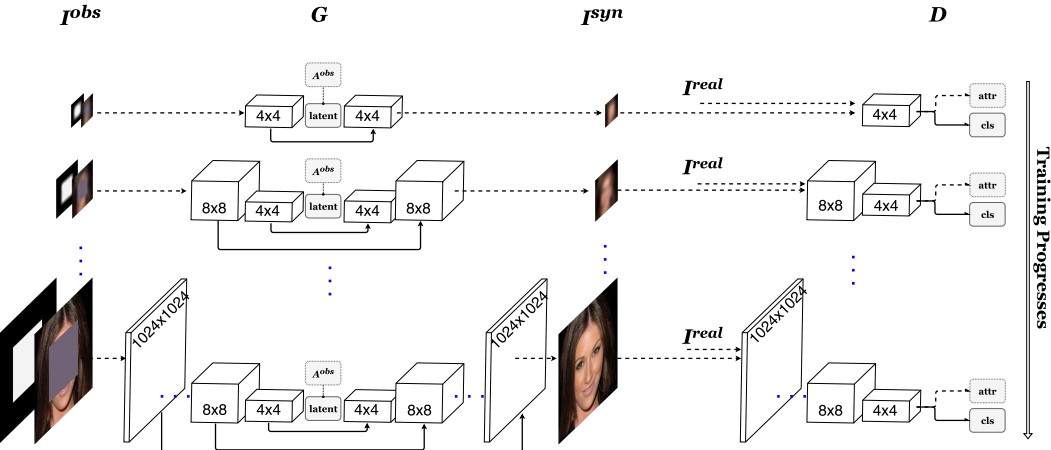

Figure 18: The progressive training process of our approach. The training of the completion network (or the "generator" $G$) and the discriminator $D$ starts at low resolution ($4 \times 4$). Higher layers are added to both $G$ and $D$ progressively to increase the resolution of the synthesized images. The $\boxed{r \times r}$ cubes in the figure represent convolutional layers that handle resolution $r$. For the conditional version, attribute labels $A^{obs}$ are concatenated to the latent vectors. The discriminator $D$ splits into two branches in the final layers: $D_{cls}$ that classifies if an input image is real, and $D_{attr}$ that predicts attribute vectors. Note that $X^G$ and $X^D$ are both a set of inputs as defined in the paper. We use images in this Figure as a simplified illustration.

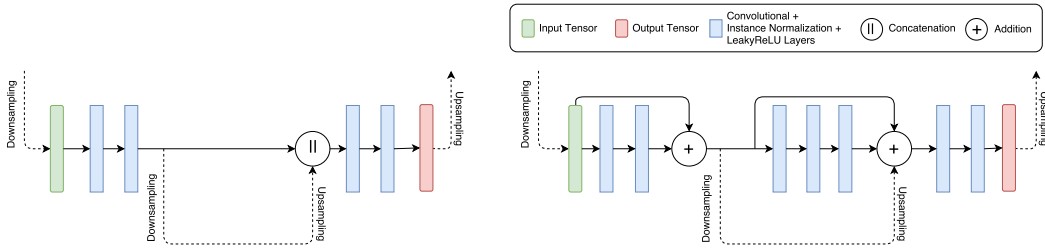

Figure 19: Illustrations of a single layer of our architecture. There are skip connections between mirrored encoder and decoder layers. Left: the structure of the completion network; the skip connection is a copy-and-concatenate operation. This structure helps preserve the identity information between the synthesized images and real faces, resulting in little deformation. Right: the structure of the conditional completion network; residual connections are added to the encoder, and the skip connections are residual blocks instead of direct concatenation. The attributes of the synthesized contents can be manipulated more easily with this structure. Each blue rectangle represents a set of Convolutional, Instance Normalization and Leaky Rectified Linear Unit (LeakyReLU) (Maas et al., 2013) layers.

Table 3: Top: the Encoding component of generator $G_{enc}$; Bottom: Latent Layer. $N$ is the length of an attribute vector. The attribute concatenation operation (AttrConcat) is only activated for our conditional model.

| Type | Kernel | Stride | Output Shape |
|---|---|---|---|
| Input Image | - | - | $4 \times 1024 \times 1024$ |
| Conv | $1 \times 1$ | $1 \times 1$ | $16 \times 1024 \times 1024$ |
| Conv | $3 \times 3$ | $1 \times 1$ | $32 \times 1024 \times 1024$ |
| Conv | $3 \times 3$ | $1 \times 1$ | $32 \times 1024 \times 1024$ |
| Downsample | - | - | $32 \times 512 \times 512$ |
| Conv | $3 \times 3$ | $1 \times 1$ | $64 \times 512 \times 512$ |
| Conv | $3 \times 3$ | $1 \times 1$ | $64 \times 512 \times 512$ |
| Downsample | - | - | $64 \times 256 \times 256$ |
| Conv | $3 \times 3$ | $1 \times 1$ | $128 \times 256 \times 256$ |
| Conv | $3 \times 3$ | $1 \times 1$ | $128 \times 256 \times 256$ |
| Downsample | - | - | $128 \times 128 \times 128$ |
| Conv | $3 \times 3$ | $1 \times 1$ | $256 \times 128 \times 128$ |
| Conv | $3 \times 3$ | $1 \times 1$ | $256 \times 128 \times 128$ |
| Downsample | - | - | $256 \times 64 \times 64$ |
| Conv | $3 \times 3$ | $1 \times 1$ | $512 \times 64 \times 64$ |
| Conv | $3 \times 3$ | $1 \times 1$ | $512 \times 64 \times 64$ |
| Downsample | - | - | $512 \times 32 \times 32$ |
| Conv | $3 \times 3$ | $1 \times 1$ | $512 \times 32 \times 32$ |
| Conv | $3 \times 3$ | $1 \times 1$ | $512 \times 32 \times 32$ |
| Downsample | - | - | $512 \times 16 \times 16$ |
| Conv | $3 \times 3$ | $1 \times 1$ | $512 \times 16 \times 16$ |
| Conv | $3 \times 3$ | $1 \times 1$ | $512 \times 16 \times 16$ |
| Downsample | - | - | $512 \times 8 \times 8$ |
| Conv | $3 \times 3$ | $1 \times 1$ | $512 \times 8 \times 8$ |
| Conv | $3 \times 3$ | $1 \times 1$ | $512 \times 8 \times 8$ |
| Downsample | - | - | $512 \times 4 \times 4$ |

| Type | Kernel | Stride | Output Shape |
|---|---|---|---|
| Conv | $3 \times 3$ | $1 \times 1$ | $512 \times 4 \times 4$ |
| Conv | $4 \times 4$ | $1 \times 1$ | $512 \times 1 \times 1$ |
| AttrConcat | optional | - | $512(+N) \times 1 \times 1$ |
| Conv | $4 \times 4$ | $1 \times 1$ | $512 \times 4 \times 4$ |
| Conv | $3 \times 3$ | $1 \times 1$ | $512 \times 4 \times 4$ |

Table 4: The completion component of generator $G_{dec}$. Depending on the particular operation of the skip connection (Skip), the number of filters is either doubled (for concatenation operations) or remains the same (for addition operations). In practice, $G_{dec}$ output a feature map that can be used to generate a RGB image (with *ToRGB* layers) or predict a read/write Filter (with *ToFilter* layers, see Table 5).

| Type | Kernel | Stride | Output Shape |
|---|---|---|---|
| Upsample | - | - | $512 \times 8 \times 8$ |
| Skip | - | - | $1024\,(512) \times 8 \times 8$ |
| Conv | $3 \times 3$ | $1 \times 1$ | $512 \times 8 \times 8$ |
| Conv | $3 \times 3$ | $1 \times 1$ | $512 \times 8 \times 8$ |
| Upsample | - | - | $512 \times 16 \times 16$ |
| Skip | - | - | $1024\,(512) \times 16 \times 16$ |
| Conv | $3 \times 3$ | $1 \times 1$ | $512 \times 16 \times 16$ |
| Conv | $3 \times 3$ | $1 \times 1$ | $512 \times 16 \times 16$ |
| Upsample | - | - | $512 \times 32 \times 32$ |
| Skip | - | - | $1024\,(512) \times 32 \times 32$ |
| Conv | $3 \times 3$ | $1 \times 1$ | $512 \times 32 \times 32$ |
| Conv | $3 \times 3$ | $1 \times 1$ | $512 \times 32 \times 32$ |
| Upsample | - | - | $512 \times 64 \times 64$ |
| Skip | - | - | $1024\,(512) \times 64 \times 64$ |
| Conv | $3 \times 3$ | $1 \times 1$ | $512 \times 64 \times 64$ |
| Conv | $3 \times 3$ | $1 \times 1$ | $512 \times 64 \times 64$ |
| Upsample | - | - | $512 \times 128 \times 128$ |
| Conv | $3 \times 3$ | $1 \times 1$ | $256 \times 128 \times 128$ |
| Skip | - | - | $512\,(256) \times 128 \times 128$ |
| Conv | $3 \times 3$ | $1 \times 1$ | $256 \times 128 \times 128$ |
| Conv | $3 \times 3$ | $1 \times 1$ | $256 \times 128 \times 128$ |
| Upsample | - | - | $256 \times 256 \times 256$ |
| Conv | $3 \times 3$ | $1 \times 1$ | $128 \times 256 \times 256$ |
| Skip | - | - | $256\,(128) \times 256 \times 256$ |
| Conv | $3 \times 3$ | $1 \times 1$ | $128 \times 256 \times 256$ |
| Conv | $3 \times 3$ | $1 \times 1$ | $128 \times 256 \times 256$ |
| Upsample | - | - | $128 \times 512 \times 512$ |
| Conv | $3 \times 3$ | $1 \times 1$ | $64 \times 512 \times 512$ |
| Skip | - | - | $128\,(64) \times 512 \times 512$ |
| Conv | $3 \times 3$ | $1 \times 1$ | $64 \times 512 \times 512$ |
| Conv | $3 \times 3$ | $1 \times 1$ | $64 \times 512 \times 512$ |
| Upsample | - | - | $64 \times 1024 \times 1024$ |
| Conv | $3 \times 3$ | $1 \times 1$ | $32 \times 1024 \times 1024$ |
| Skip | - | - | $64\,(32) \times 1024 \times 1024$ |

Table 5: Left: The *ToRGB* layers that convert feature maps to RGB images. Right: *ToFilter* layers that predict a read/write filter from feature maps.

| Conv | $3 \times 3$ | $1 \times 1$ | $32 \times 1024 \times 1024$ |
|---|---|---|---|
| Conv | $3 \times 3$ | $1 \times 1$ | $32 \times 1024 \times 1024$ |
| Conv | $1 \times 1$ | $1 \times 1$ | $3 \times 1024 \times 1024$ |

| Conv | $3 \times 3$ | $1 \times 1$ | $64 \times 1024 \times 1024$ |
|---|---|---|---|
| Conv | $3 \times 3$ | $1 \times 1$ | $64 \times 1024 \times 1024$ |
| Conv | $1 \times 1$ | $1 \times 1$ | $1 \times 1024 \times 1024$ |

Table 6: Top: Feature Network $\mathbb{F}(\cdot)$ computes a feature map for an input image, which is later used by $D_{\text{cls}}$ and $D_{\text{attr}}$; Middle: The real/fake head classifier $D_{\text{cls}}$; Bottom: The attribute network $D_{\text{attr}}$. $N$ is the length of an attribute vector. This network is only activated for the conditional model.

| Type | Kernel | Stride | Output Shape |
|---|---|---|---|
| Input Image | - | - | $3 \times 1024 \times 1024$ |
| Conv | $1 \times 1$ | $1 \times 1$ | $16 \times 1024 \times 1024$ |
| Conv | $3 \times 3$ | $1 \times 1$ | $16 \times 1024 \times 1024$ |
| Conv | $3 \times 3$ | $1 \times 1$ | $32 \times 1024 \times 1024$ |
| Downsample | - | - | $32 \times 512 \times 512$ |
| Conv | $3 \times 3$ | $1 \times 1$ | $32 \times 512 \times 512$ |
| Conv | $3 \times 3$ | $1 \times 1$ | $64 \times 512 \times 512$ |
| Downsample | - | - | $64 \times 256 \times 256$ |
| Conv | $3 \times 3$ | $1 \times 1$ | $64 \times 256 \times 256$ |
| Conv | $3 \times 3$ | $1 \times 1$ | $128 \times 256 \times 256$ |
| Downsample | - | - | $128 \times 128 \times 128$ |
| Conv | $3 \times 3$ | $1 \times 1$ | $128 \times 128 \times 128$ |
| Conv | $3 \times 3$ | $1 \times 1$ | $256 \times 128 \times 128$ |
| Downsample | - | - | $256 \times 64 \times 64$ |
| Conv | $3 \times 3$ | $1 \times 1$ | $256 \times 64 \times 64$ |
| Conv | $3 \times 3$ | $1 \times 1$ | $512 \times 64 \times 64$ |
| Downsample | - | - | $512 \times 32 \times 32$ |
| Conv | $3 \times 3$ | $1 \times 1$ | $512 \times 32 \times 32$ |
| Conv | $3 \times 3$ | $1 \times 1$ | $512 \times 32 \times 32$ |
| Downsample | - | - | $512 \times 16 \times 16$ |
| Conv | $3 \times 3$ | $1 \times 1$ | $512 \times 16 \times 16$ |
| Conv | $3 \times 3$ | $1 \times 1$ | $512 \times 16 \times 16$ |
| Downsample | - | - | $512 \times 8 \times 8$ |
| Conv | $3 \times 3$ | $1 \times 1$ | $512 \times 8 \times 8$ |
| Conv | $3 \times 3$ | $1 \times 1$ | $512 \times 8 \times 8$ |
| Downsample | - | - | $512 \times 4 \times 4$ |

| Type | Kernel | Stride | Output Shape |
|---|---|---|---|
| Conv | $3 \times 3$ | $1 \times 1$ | $512 \times 4 \times 4$ |
| Conv | $4 \times 4$ | $1 \times 1$ | $1 \times 1 \times 1$ |

| Type | Kernel | Stride | Output Shape |
|---|---|---|---|
| Conv | $3 \times 3$ | $1 \times 1$ | $512 \times 4 \times 4$ |
| Conv | $4 \times 4$ | $1 \times 1$ | $N \times 1 \times 1$ |

