# OpenReview forum: "Towards Controllable and Interpretable Face Completion via  Structure-Aware and Frequency-Oriented Attentive GANs"
_ICLR.cc/2020/Conference — Reject_

### Official Review · AnonReviewer1 · 2019-10-23
**Official Blind Review #1**

**Rating:** 6

**Review:**

This paper proposes a face completion network that synthesizes the missing part in the face images with GANs. Using facial landmarks and facial attributes, face completion became controllable as both are used as conditional information (input) for the generation (synthesis). Moreover, the proposed Frequency-Oriented Attention Module (FOAM) enables an interpretable coarse-to-fine progressive generative process. The proposed methods show significant improvement in the completion quality.

Overall,  the method shows how the face completion can be controlled and how the face completion is done by improving details. The attentive framework makes possible to do kinds of band-pass filtering. The results are impressive but have some concerns as the following:

- Have you tried to train models without using facial landmarks? Are facial landmarks only for controlling facial expressions?
- As I carefully looked at the generated faces, many of them have asymmetric (unbalanced) eyes. Is it due to the predicted facial landmarks?
- Is there any randomness (random input) involved during completion (generation)? Is this model possible to generate different faces from the same conditional input?
- What happens if the attributes are interpolated rather than zero or one?
- The order of Fig 4 and 5 seems weird.

**Experience Assessment:**

I have published one or two papers in this area.

**Review Assessment: Checking Correctness Of Derivations And Theory:**

I assessed the sensibility of the derivations and theory.

**Review Assessment: Checking Correctness Of Experiments:**

I assessed the sensibility of the experiments.

**Review Assessment: Thoroughness In Paper Reading:**

I read the paper at least twice and used my best judgement in assessing the paper.

---

> ### Author Response · Authors · 2019-11-15
> **Response to Reviewer #1**
>
> We thank the reviewer for the comments and appreciation, and would like to answer the reviewer’s concerns as follows:
>
> Q1: Have you tried to train models without using facial landmarks? Are facial landmarks only for controlling facial expressions?
> Yes, we have tried this. It performed worse on face completion tasks without the facial landmarks, especially when the masks were irregular (e.g. second and third images in Figure 6). The predicted facial landmarks (in our current pipeline) provide high level presentations of face structures, which helps the generation process, and enable the control of facial expressions.
>
> Q2:  As I carefully looked at the generated faces, many of them have asymmetric (unbalanced) eyes. Is it due to the predicted facial landmarks?
> That is right, because the generation process is conditioned on these predicted landmarks. In this work, we have not specifically dealt with the symmetric issue. It will be an interesting topic  for future work, e.g., by introducing a symmetry-induced loss function and/or leveraging more powerful off-the-shelf facial landmark detectors.
>
> Q3: Is there any randomness (random input) involved during completion (generation)? Is this model possible to generate different faces from the same conditional input? &  What happens if the attributes are interpolated rather than zero or one?
> For the model in this paper, there is no randomness involved. But we made a video demo (not included in this submission) to show the synthesized faces from interpolated attributes (some snapshots of the demo video is added as Figure 15 in the appendix). All of the faces are very natural, and the transitions between attributes (e.g. from smile to not smile, from male to female) are very smooth. It means this model is able to produce a variety of images. The results of soft attribute control will be added into result figures. Randomness can be achieved by attaching a random noise vector to the attribute vector.
>
> Q4: The order of Fig 4 and 5 seems weird.
> We fixed the layout issue in the revision.

---

### Official Review · AnonReviewer2 · 2019-10-23
**Official Blind Review #2**

**Rating:** 3

**Review:**

This paper proposes controllable and interpretable high-resolution and fast face completion by learning generative adversarial networks (GANs) progressively from low resolution to high resolution. It combines the masks, landmarks, corrupted images as inputs to generate completed images in high-resolution. The proposed frequency-oriented attentive module (FOAM) encourages GANs to highlight much more to finer details in the coarse-to-fine progressive training, thus enabling progressive attention to face structures.

Integrating the Progressive growing GAN (PGGAN) for high-resolution face completion is an interesting step-up work after the success of PGGAN on high-resolution image generation.

Basically the FOAM is proposed to merge the images from different resolution levels instead of a weighted summation. The novelty, however, is not strong as the simple summation provides reasonable performance as convinced in the progressive growing GAN paper. And there lacks quantitative analysis on ablation study as showed in Figure 4. Therefore, the contributions of each component are not convincing.

Using Progressive Growing GAN for high-resolution face completion is also studied in [1]. Figure 17, Figure 18, indicates the author used the same structure as used by [1]. However,  [1] is not cited and there is no comparison between these two models in the paper.

[1] Zeyuan Liu, et al  "High Resolution Face Completion with Multiple Controllable Attributes via Fully End-to-End Progressive Generative Adversarial Networks "

**Experience Assessment:**

I have read many papers in this area.

**Review Assessment: Checking Correctness Of Derivations And Theory:**

I assessed the sensibility of the derivations and theory.

**Review Assessment: Checking Correctness Of Experiments:**

I assessed the sensibility of the experiments.

**Review Assessment: Thoroughness In Paper Reading:**

I read the paper at least twice and used my best judgement in assessing the paper.

---

> ### Author Response · Authors · 2019-11-15
> **Response to Reviewer #2**
>
> We thank the reviewer for the comments and appreciation, and would like to answer the reviewer’s concerns as follows:
>
> Q1:  The novelty, however, is not strong as the simple summation provides reasonable performance as convinced in the progressive growing GAN paper. And there lacks quantitative analysis on ablation study as showed in Figure 4. Therefore, the contributions of each component are not convincing.
> We would like to point out that the convincing performance by the simple summation in PGGAN is for general unconditional image synthesis, not necessarily  transferrable to the face completion problem. Our ablation study qualitatively supports the necessity of the proposed FOAM, as well as our intuitive analyses after Eqn.3 (page 4). That being said, we agree that it will be better to do a quantitative study in addition to Figure 4.  One challenge in evaluating face completion is that there is no well-accepted metric at present. We conduct a pilot human study for quantitative evaluation, but mainly focus on comparisons between the proposed method and previous state-of-the-art method due to the time-consuming setup of human study. Furthermore, based on the qualitative results shown in Figure 4, we suspect that if performed, the human study may provide consistent evaluation results on the proposed method.

---

### Official Review · AnonReviewer3 · 2019-11-03
**Official Blind Review #141**

**Rating:** 3

**Review:**

This paper aims at the problem of face synthesis. The authors propose a progressive GAN with frequency-oriented attention modules for high resolution and fast controllable and interpretable face completion, which learns face structures from coarse to fine guided by the FOAM. Experiments are conducted to verify the effectiveness of the proposed method. This paper is well written and is easy to understand.

1. The most interesting idea is the frequency-oriented attention modules ,while the idea of structure-aware seems very common in the area.

2. The experiments are unconvincing. There are so many works about face synthesis in recent years. Why do the authors only compare with GL and CTX? Also, the authors do not study how each component affects the final performance, which is very important for the reader to understand why it works.

3. In general, the framework of the proposed method is very common. This paper only follow previous work and lacks new insights about the problem.

**Experience Assessment:**

I do not know much about this area.

**Review Assessment: Checking Correctness Of Derivations And Theory:**

I assessed the sensibility of the derivations and theory.

**Review Assessment: Checking Correctness Of Experiments:**

I carefully checked the experiments.

**Review Assessment: Thoroughness In Paper Reading:**

I read the paper thoroughly.

---

> ### Author Response · Authors · 2019-11-15
> **Response to Reviewer #141**
>
> We thank the reviewer for the comments and appreciation, and would like to answer the reviewer’s concerns as follows:
>
> Q1: "There are so many works about face synthesis in recent years. Why do the authors only compare with GL and CTX?"
> We study face completion with arbitrary masks in this paper, which is a challenging conditional face synthesis, and different from general unconditional face synthesis. GL and CTX are two of the previous state-of-the-art face completion methods.  And, due to the lack of quantitative metrics for image completion (Yeh et al., Yu et al.), we resort to  evaluate these methods with a “gold standard”, i.e., the human study and careful statistical analyses (Figure 9 and appendix A.2). In the human pilot study, it is more accurate and feasible for the observers to compare the pairs of images (e.g. ours vs. CTX or real vs. synthesized). We can already see our method outperformed state-of-the-art method (CTX) significantly.
>
> Q2: "the authors do not study how each component affects the final performance, which is very important for the reader to understand why it works."
> We provide an ablation study (Figure 4) which qualitatively shows the impact of essential  components (the proposed FOAM and the well-executed loss functions)  of  our  method.
>
> Q3: "the framework of the proposed method is very common. This paper only follow previous work and lacks new insights about the problem. "
> We agree that the overall proposed framework is common, which is based on progressive GANs, one of the most widely used framework in generative learning. But, we would like to point out that we do not only follow previous work. We present a novel FOAM module that are critical for interpretable and controllable high-resolution face completion without any post-processing. The proposed FOAM addresses the common stability issue in training progressive GANs (as compared in the ablation study, Figure 4). We propose a well-executed set of loss functions which are non-trivial for face completion with arbitrary masks. We also obtain state-of-the-art performance.

---

### Decision · Program_Chairs · 2019-12-19

**Decision:**

Reject

**Comment:**

This work performs fast controllable and interpretable face completion, by proposing a progressive GAN with frequency-oriented attention modules (FOAM).  The proposed FOAM encourages GANs to highlight more to finer details in the progressive training process. This paper is well written and is easy to understand. While reviewer #1 is overall positive about this work, the reviewer #2 and #141 rated weak reject with various concerns, including unconvincing experiments, very common framework, limited novelty, and the lack of ablation study. The authors provided response to the questions, but did not change the rating of the reviewers. Given the various concerns raised, the ACs agree that this paper can not be accepted at its current state.